# COUNT-BASED EXPLORATION WITH THE SUCCESSOR REPRESENTATION

## ABSTRACT

The problem of exploration in reinforcement learning is well-understood in the tabular case and many sample-efficient algorithms are known. Nevertheless, it is often unclear how the algorithms in the tabular setting can be extended to tasks with large state-spaces where generalization is required. Recent promising developments generally depend on problem-specific density models or handcrafted features. In this paper we introduce a simple approach for exploration that allows us to develop theoretically justified algorithms in the tabular case but that also give us intuitions for new algorithms applicable to settings where function approximation is required. Our approach and its underlying theory is based on the substochastic successor representation, a concept we develop here. While the traditional successor representation is a representation that defines state generalization by the similarity of successor states, the substochastic successor representation is also able to implicitly count the number of times each state (or feature) has been observed. This extension connects two until now disjoint areas of research. We show in traditional tabular domains (RiverSwim and SixArms) that our algorithm empirically performs as well as other sample-efficient algorithms. We then describe a deep reinforcement learning algorithm inspired by these ideas and show that it matches the performance of recent pseudo-count-based methods in hard exploration Atari 2600 games.

## 1 INTRODUCTION

Reinforcement learning (RL) tackles sequential decision making problems by formulating them as tasks where an agent must learn how to act optimally through trial and error interactions with the environment. The goal in these problems is to maximize the sum of the numerical reward signal observed at each time step. Because the actions taken by the agent influence not just the immediate reward but also the states and associated rewards in the future, sequential decision making problems require agents to deal with the trade-off between immediate and delayed rewards. Here we focus on the problem of exploration in RL, which aims to reduce the number of samples (i.e., interactions) an agent needs in order to learn to perform well in these tasks when the environment is initially unknown.

The sample efficiency of RL algorithms is largely dependent on how agents select exploratory actions. In order to learn the proper balance between immediate and delayed rewards agents need to navigate through the state space to learn about the outcome of different transitions. The number of samples an agent requires is related to how quickly it is able to explore the state-space. Surprisingly, the most common approach is to select exploratory actions uniformly at random, even in high-profile success stories of RL (e.g., Tesauro, 1995; Mnih et al., 2015). Nevertheless, random exploration often fails in environments with sparse rewards, that is, environments where the agent observes a reward signal of value zero for the majority of states.[1]

In model-based approaches agents explicitly learn a model of the dynamics of the environment which they use to plan future actions. In this setting the problem of exploration is well understood. When all states can be enumerated and uniquely identified (tabular case), we have algorithms with proven sample complexity bounds on the maximum number of suboptimal actions an agent selects before

---

[1]When we refer to environments with sparse rewards we do so for brevity and ease of presentation. Actually, any sequential decision making problem has dense rewards. In the RL formulation a reward signal is observed at every time step. By environments with sparse rewards we mean environments where the vast majority of transitions lead to reward signals with the same value.

converging to an $\epsilon$-optimal policy (e.g., Brafman & Tennenholtz, 2002; Kearns & Singh, 2002; Strehl & Littman, 2008). However, these approaches are not easily extended to large environments where it is intractable to enumerate all of the states. When using function approximation, the concept of state visitation is not helpful and learning useful models is by itself quite challenging.

Due to the difficulties in learning good models in large domains, model-free methods are much more popular. Instead of building an explicit model of the environment, they estimate state values directly from transition samples (state, action, reward, next state). Unfortunately, this approach makes systematic exploration much more challenging. Nevertheless, because model-free methods make up the majority of approaches scalable to large domains, practitioners often ignore the exploration challenges these methods pose and accept the high sample complexity of random exploration. Reward bonuses that promote exploration are one alternative to random walks (e.g., Bellemare et al., 2016; Martin et al., 2017), but none such proposed solutions are widely adopted in the field.

In this paper we introduce an algorithm for exploration based on the successor representation (SR). The SR, originally introduced by Dayan (1993), is a representation that generalizes between states using the similarity between their successors, i.e., the similarity between the states that follow the current state given the environment's dynamics and the agent's policy. The SR is defined for any problem, it can be learned through temporal-difference learning (Sutton, 1988) and, as we discuss below, it can also be seen as implicitly estimating the transition dynamics of the environment. Our approach is inspired by the substochastic successor representation (SSR), a concept we introduce here. The SSR is defined so that it implicitly counts state visitation, allowing us to use it to encourage exploration. This idea connects representation learning and exploration, two otherwise disjoint areas of research. The SSR allows us to derive an exploration bonus that when applied to model-based RL generates algorithms that perform as well as theoretically sample-efficient algorithms. Importantly, the intuition developed with the SSR assists us in the design of a model-free deep RL algorithm that achieves performance similar to pseudo-count-based methods in hard exploration Atari 2600 games (Bellemare et al., 2016; Ostrovski et al., 2017).

## 2 PRELIMINARIES

We consider an agent interacting with its environment in a sequential manner. Starting from a state $S_0 \in \mathcal{S}$, at each step the agent takes an action $A_t \in \mathcal{A}$, to which the environment responds with a state $S_{t+1} \in \mathcal{S}$ according to a transition probability function $p(s'|s,a) = \Pr(S_{t+1} = s'|S_t = s, A_t = a)$, and with a reward signal $R_{t+1} \in \mathbb{R}$, where $r(s,a)$ indicates the expected reward for a transition from state $s$ under action $a$, that is, $r(s,a) \doteq \mathbb{E}[R_t|S_t = s, A_t = a]$.

The value of a state $s$ when following a policy $\pi$, $v_\pi(s)$, is defined to be the expected sum of discounted rewards from that state: $v_\pi(s) \doteq \mathbb{E}_\pi\left[\sum_{k=t+1}^{T}\gamma^{k-t-1}R_k\Big|S_t = s\right]$, with $\gamma$ being the discount factor. When the transition probability function $p$ and the reward function $r$ are known, we can compute $v_\pi(s)$ recursively by solving the system of equations below (Bellman, 1957):

$$v_\pi(s) = \sum_a \pi(a|s)\big[r(s,a) + \gamma \sum_{s'} p(s'|s,a)v_\pi(s')\big].$$

This equation can also be written in matrix form with $\mathbf{v}_\pi, \mathbf{r} \in \mathbb{R}^{|\mathcal{S}|}$ and $P_\pi \in \mathbb{R}^{|\mathcal{S}|\times|\mathcal{S}|}$:

$$\mathbf{v}_\pi = \mathbf{r} + \gamma P_\pi \mathbf{v}_\pi = (I - \gamma P_\pi)^{-1}\mathbf{r}, \tag{1}$$

where $P_\pi$ is the state to state transition probability function induced by $\pi$, that is, $P_\pi(s,s') = \sum_a \pi(a|s)p(s'|s,a)$.

Traditional model-based algorithms for RL work by learning estimates of the matrix $P_\pi$ and of the vector $\mathbf{r}$ and using them to estimate $v_\pi$, for example by solving Equation 1. We use $\hat{P}_\pi$ and $\hat{\mathbf{r}}$ to denote empirical estimates of $P_\pi$ and $\mathbf{r}$. Formally,

$$\hat{P}_\pi(s'|s) = \frac{n(s,s')}{n(s)}, \qquad\qquad \hat{\mathbf{r}}(s) = \frac{C(s,s')}{n(s)}, \tag{2}$$

where $\hat{\mathbf{r}}(i)$ denotes the $i$-th entry in the vector $\hat{\mathbf{r}}$, $n(s,s')$ is the number of times the transition $s \to s'$ was observed, $n(s) = \sum_{s' \in \mathcal{S}} n(s,s')$, and $C(s,s')$ is the sum of the rewards associated with the $n(s,s')$ transitions (we drop the action in the discussion to simplify notation).

Alternatively, in model-free RL, instead of estimating $P_\pi$ and $\mathbf{r}$ we estimate $v_\pi(s)$ directly from samples. We often use temporal-difference (TD) learning (Sutton, 1988) to update our estimates of $v_\pi(s)$, $\hat{v}(\cdot)$, online:

$$\hat{v}(S_t) \leftarrow \hat{v}(S_t) + \alpha\big[R_{t+1} + \gamma\hat{v}(S_{t+1}) - \hat{v}(S_t)\big], \tag{3}$$

where $\alpha$ is the step-size parameter. Generalization is required in problems with large state spaces, where it is unfeasible to learn an individual value for each state. We do so by parametrizing $\hat{v}(s)$ with a set of weights $\theta$. We write, given the weights $\theta$, $\hat{v}(s;\theta) \approx v_\pi(s)$ and $\hat{q}(s,a;\theta) \approx q_\pi(s,a)$, where $q_\pi(s,a) = r(s,a) + \gamma\sum_{s'} p(s'|s,a)v_\pi(s')$. Model-free methods have performed well in problems with large state spaces, mainly due to the use of neural networks as function approximators (e.g., Mnih et al., 2015).

Our algorithm is based on the successor representation (SR; Dayan, 1993). The successor representation, with respect to a policy $\pi$, $\Psi_\pi$, is defined as

$$\Psi_\pi(s, s') = \mathbb{E}_{\pi,p}\Big[\sum_{t=0}^\infty \gamma^t \mathbb{I}\{S_t = s'\}\Big| S_0 = s\Big],$$

where we assume the sum is convergent with $\mathbb{I}$ denoting the indicator function. Dayan (1993) has shown that this expectation can be estimated from samples through TD learning. It also corresponds to the Neumann series of $\gamma P$:

$$\Psi_\pi = \sum_{t=0}^\infty \gamma^t (P_\pi)^t = (I - \gamma P_\pi)^{-1}. \tag{4}$$

Notice that the SR is part of the solution when computing a value function: $\mathbf{v}_\pi = \Psi_\pi \mathbf{r}$ (Equation 1). We use $\hat{\Psi}_\pi$ to denote the SR computed through $\hat{P}_\pi$, the approximation of $P_\pi$.

The definition of the SR can also be extended to features. Successor features generalize the SR to the function approximation setting (Barreto et al., 2017). We use the definition for the uncontrolled case in this paper. Importantly, the successor features can also be learned with TD learning.

**Definition 2.1** (Successor Features). *For a given $0 \le \gamma < 1$, policy $\pi$, and for a feature representation $\phi(s) \in \mathbb{R}^d$, the successor features for a state $s$ are:*

$$\boldsymbol{\psi}_\pi(s) = \mathbb{E}_{\pi,p}\left[\sum_{t=0}^\infty \gamma^t \boldsymbol{\phi}(S_t)\Big| S_0 = s\right].$$

Alternatively, in matrix form, $\Psi_\pi = \sum_{t=0}^\infty \gamma^t (P_\pi)^t \Phi = (I - \gamma P_\pi)^{-1}\Phi$. Notice that this definition reduces to the SR in the tabular case, where $\Phi = I$.

## 3   THE SUBSTOCHASTIC SUCCESSOR REPRESENTATION

In this section we introduce the concept of the *substochastic successor representation* (SSR). The SSR is derived from an empirical transition matrix similar to Equation 2, but where each state incorporates a small $(1/(n(s)+1))$ probability of terminating at that state, rather than transiting to a next state. As we will show, we can recover the visit counts $n(s)$ through algebraic manipulation on the SSR.

While computing the SSR is usually impractical, we use it as inspiration in the design of a new deep reinforcement learning algorithm for exploration (Section 4). In a nutshell, we view the SSR as approximating the process of learning the SR from an uninformative initialization (i.e., the zero vector), and using a stochastic update rule. While this approximation is relatively coarse, we believe it gives qualitative justification to our use of the learned SR to guide exploration. To further this claim, we demonstrate that using the SSR in synthetic, tabular settings yields comparable performance to that of theoretically-derived exploration algorithms.

**Definition 3.1** (Substochastic Successor Representation). *Let $\tilde{P}_\pi$ denote the substochastic matrix induced by the environment's dynamics and by the policy $\pi$ such that $\tilde{P}_\pi(s'|s) = \frac{n(s,s')}{n(s)+1}$. For a given $0 \le \gamma < 1$, the substochastic successor representation, $\tilde{\Psi}_\pi$, is defined as:*

$$\tilde{\Psi}_\pi = \sum_{t=0}^\infty \gamma^t \tilde{P}_\pi{}^t = (I - \gamma\tilde{P}_\pi)^{-1}.$$

The theorem below formalizes the idea that the $\ell_1$ norm of the SSR implicitly counts state visitation.

**Theorem 1.** *Let $n(s)$ denote the number of times state $s$ has been visited and let $\chi(s) = (1 + \gamma) - ||\tilde{\Psi}_\pi(s)||_1$, where $\tilde{\Psi}_\pi$ is the substochastic SR as in Definition 3.1. For a given $0 \le \gamma < 1$,*

$$\frac{\gamma}{n(s) + 1} - \frac{\gamma^2}{1 - \gamma} \le \chi(s) \le \frac{\gamma}{n(s) + 1}$$

*Proof of Theorem 1.* Let $\hat{P}_\pi$ be the empirical transition matrix. We first rewrite $\tilde{P}_\pi$ in terms of $\hat{P}_\pi$:

$$\tilde{P}_\pi(s, s') = \frac{n(s, s')}{n(s) + 1} = \frac{n(s)}{n(s) + 1} \frac{n(s, s')}{n(s)} = \frac{n(s)}{n(s) + 1} \hat{P}_\pi(s, s') = \left(1 - \frac{1}{n(s) + 1}\right) \hat{P}_\pi(s, s').$$

The expression above can also be written in matrix form: $\tilde{P}_\pi = (I - N)\hat{P}_\pi$, where $N \in \mathbb{R}^{|S| \times |S|}$ denotes the diagonal matrix of augmented inverse counts. Expanding $\tilde{\Psi}_\pi$ we have:

$$\tilde{\Psi}_\pi = \sum_{t=0}^{\gamma} (\gamma \tilde{P}_\pi)^t = I + \gamma \tilde{P}_\pi + \sum_{t=2}^{\infty} (\gamma \tilde{P}_\pi)^t = I + \gamma \tilde{P}_\pi + \gamma^2 \tilde{P}_\pi^2 \tilde{\Psi}_\pi.$$

The top eigenvector of a stochastic matrix is the all-ones vector, $\mathbf{e}$ (Meyn & Tweedie, 2012), and it corresponds to the eigenvalue 1. Using this fact and the definition of $\tilde{P}_\pi$ with respect to $\hat{P}_\pi$ we have:

$$
\begin{aligned}
(I + \gamma \tilde{P}_\pi)\mathbf{e} + \gamma^2 \tilde{P}_\pi^2 \tilde{\Psi}_\pi \mathbf{e} &= \left(I + \gamma(I - N)\hat{P}_\pi\right)\mathbf{e} + \gamma^2 \tilde{P}_\pi^2 \tilde{\Psi}_\pi \mathbf{e} \\
&= (I + \gamma)\mathbf{e} - \gamma N \mathbf{e} + \gamma^2 \tilde{P}_\pi^2 \tilde{\Psi}_\pi \mathbf{e}.
\end{aligned}
\tag{5}
$$

We can now bound the term $\gamma^2 \tilde{P}_\pi^2 \tilde{\Psi}_\pi \mathbf{e}$ using the fact that $\mathbf{e}$ is also the top eigenvector of the successor representation and has eigenvalue $\frac{1}{1-\gamma}$ (Machado et al., 2018b):

$$0 \le \gamma^2 \tilde{P}_\pi^2 \tilde{\Psi}_\pi \mathbf{e} \le \frac{\gamma^2}{1 - \gamma} \mathbf{e}.$$

Plugging (5) into the definition of $\chi$ we have (notice that $\Psi(s)\mathbf{e} = ||\Psi(s)||_1$):

$$\chi(s) = (1 + \gamma)\mathbf{e} - (1 + \gamma)\mathbf{e} + \gamma N \mathbf{e} - \gamma^2 \tilde{P}_\pi^2 \tilde{\Psi}_\pi \mathbf{e} = \gamma N \mathbf{e} - \gamma^2 \tilde{P}_\pi^2 \tilde{\Psi}_\pi \mathbf{e} \le \gamma N \mathbf{e}.$$

When we also use the other bound on the quadratic term we conclude that, for any state $s$,

$$\frac{\gamma}{n(s) + 1} - \frac{\gamma^2}{1 - \gamma} \le \chi(s) \le \frac{\gamma}{n(s) + 1}.$$

$\square$

In other words, the SSR, obtained after a slight change to the SR, can be used to recover state visitation counts. The intuition behind this result is that the phantom transition, represented by the $+1$ in the denominator of the SSR, serves as a proxy for the uncertainty about that state by underestimating the SR. This is due to the fact that $\sum_{s'} \tilde{P}_\pi(s, s')$ gets closer to 1 each time state $s$ is visited.

This result can now be used to convert the SSR into a reward function in the tabular case. We do so by using the SSR to define an exploration bonus, $r_{\text{int}}$, such that the reward being maximized by the agent becomes $r(s, a) + \beta r_{\text{int}}(s)$, where $\beta$ is a scaling parameter. Since we want to incentivize agents to visit the least visited states as quickly as possible, we can trivially define $\mathbf{r}_{\text{int}} = -||\tilde{\Psi}_\pi(s)||_1$, where we penalize the agent by visiting the states that lead to commonly visited states. Notice that the shift $(1 + \gamma)$ in $\chi(s)$ has no effect as an exploration bonus because it is the same across all states.

Table 1: Comparison between our algorithm, termed ESSR, and R-MAX, $E^3$, and MBIE. The numbers reported for R-MAX, $E^3$, and MBIE are an estimate obtained from the histograms presented by Strehl & Littman (2008). The performance of our algorithm is the average over 100 runs. A 95% confidence interval is reported between parentheses.

| | $E^3$ | R-MAX | MBIE | ESSR | |
|---|---|---|---|---|---|
| RIVERSWIM | 3,000,000 | 3,000,000 | 3,250,000 | 3,088,924 | ($\pm$ 57,584) |
| SIXARMS | 1,800,000 | 2,800,000 | 9,250,000 | 7,327,222 | ($\pm$ 1,189,460) |

EVALUATING $-||\tilde{\Psi}_\pi(s)||_1$ AS AN EXPLORATION BONUS

We evaluated the effectiveness of the proposed exploration bonus in a standard model-based algorithm. In our implementation the agent updates its transition probability model and reward model through Equation 2 and its SSR estimate as in Definition 3.1 (the pseudo-code of this algorithm is available in the Appendix), which is then used for the exploration bonus $r_{\text{int}}$. We used the domains RiverSwim and SixArms (Strehl & Littman, 2008) to assess the performance of this algorithm.[2] These are traditional domains in the PAC-MDP literature (Kakade, 2003) and are often used to evaluate provably sample-efficient algorithms. Details about these environments are also available in the Appendix. We used the same protocol used by Strehl & Littman (2008). Our results are available in Table 1. It is interesting to see that our algorithm performs as well as R-MAX (Brafman & Tennenholtz, 2002) and $E^3$ (Kearns & Singh, 2002) on RiverSwim and it clearly outperforms these algorithms on SixArms.

## 4 COUNTING FEATURE ACTIVATIONS WITH THE SR

In large environments, where enumerating all states is not an option, directly using the SSR as described in the previous section is not viable. Learning the SSR becomes even more challenging when the representation, $\phi(\cdot)$, is also being learned and so is non-stationary. In this section we design an algorithm for the function approximation setting *inspired* by the results from the previous section.

Since explicitly estimating the transition probability function is not an option, we learn the SR directly using TD learning. In order to capture the SSR we rely on TD's tendency to underestimate values when the estimates are pessimistically initialized, just as the SSR underestimates the true successor representation; with larger underestimates for states (and similarly features) that are rarely observed. This is mainly due to the fact that when the SR is being learned with TD learning, because a reward of 1 is observed at each time step, there is no variance in the target and the predictions slowly approach the true value of the SR. When pessimistically initialized, the predictions approach the target from below. In this sense, what defines how far a prediction is from its final target is indeed how many times it has been updated in a given state. Finally, recent work (Kulkarni et al., 2016; Machado et al., 2018b) have shown successor features can be learned jointly with the feature representation itself. These ideas are combined together to create our algorithm.

The neural network we used to learn the agent's value function while also learning the feature representation and the successor representation is depicted in Figure 1. The layers used to compute the state-action value function, $\hat{q}(S_t, \cdot)$, are structured as in DQN (Mnih et al., 2015), but with different numbers of parameters (i..e, filter sizes, stride, and number of nodes). This was done to match Oh et al.'s (2015) architecture, which is known to succeed in the auxiliary task we define below. From here on, we will call the part of our architecture that predicts $\hat{q}(S_t, \cdot)$ DQN$_e$. It is trained to minimize

$$\mathcal{L}_{\text{TD}} = \mathbb{E}\Big[\big((1-\tau)\delta(s,a) + \tau\delta_{\text{MC}}(s,a)\big)^2\Big],$$

---

[2]Our algorithm maximizes the discounted return ($\gamma = 0.95$). We used policy iteration where the policy evaluation step is terminated when the estimates of the value function change by less than 0.01. In RiverSwim $\beta$ was set to 100 and in SixArms $\beta$ was set to 1000. These values were obtained after evaluating the algorithm for $\beta \in \{1, 10, 100, 200, 1000, 2000\}$. The code used to generate these results is available at: `https://github.com/ommitted/blind`.

Figure 1: Neural network architecture used by our algorithm when learning to play Atari 2600 games.

where $\delta(s,a)$ and $\delta_{\text{MC}}(s,a)$ are defined as

$$\delta(s,a) = R_t + \beta r_{\text{int}}(s;\theta^-) + \gamma \max_{a'} q(s',a';\theta^-) - q(s,a;\theta),$$

$$\delta_{\text{MC}}(s,a) = \sum_{t=0}^{\infty} \gamma^t \Big( r(S_t, A_t) + \beta r_{\text{int}}(S_t;\theta^-) \Big) - q(s,a;\theta).$$

This loss is known as the mixed Monte-Carlo return (MMC) and it has been used in the past by the algorithms that achieved succesful exploration in deep reinforcement learning (Bellemare et al., 2016; Ostrovski et al., 2017). The distinction between $\theta$ and $\theta^-$ is standard in the field, with $\theta^-$ denoting the parameters of the target network, which is updated less often for stability purposes (Mnih et al., 2015). As before, we use $r_{\text{int}}$ to denote the exploration bonus obtained from the successor features of the internal representation, $\phi(\cdot)$, which will be defined below. Moreover, to ensure all features are in the same range, we normalize the feature vector so that $||\phi(\cdot)||_2 = 1$. In Figure 1 we highlight the layer in which we normalize its output with the symbol $\phi$. Notice that the features are always non-negative due to the use of ReLU gates.

The successor features are computed by the two bottom layers of the network, which minimize the loss

$$\mathcal{L}_{\text{SR}} = \mathbb{E}_{\pi,p}\Big[ \big( \phi(S_t;\theta^-) + \gamma\psi(S_{t+1};\theta^-) - \psi(S_t;\theta) \big)^2 \Big].$$

Zero is a fixed point for the SR. This is particularly concerning in settings with sparse rewards. The agent might learn to set $\phi(\cdot) = \vec{0}$ to achieve zero loss. We address this problem by not propagating $\nabla\mathcal{L}_{\text{SR}}$ to $\phi(\cdot)$ (this is depicted in Figure 1 as an open circle stopping the gradient), and by creating an auxiliary task (Jaderberg et al., 2017) to encourage a representation to be learned before a non-zero reward is observed. As Machado et al. (2018b), we use the auxiliary task of predicting the next observation, learned through the architecture proposed by Oh et al. (2015), which is depicted as the top layers in Figure 1. The loss we minimize for this last part of the network is $\mathcal{L}_{\text{Recons}} = \big( \hat{S}_{t+1} - S_{t+1} \big)^2$.

The overall loss minimized by the network is $\mathcal{L} = w_{\text{TD}}\mathcal{L}_{\text{TD}} + w_{\text{SR}}\mathcal{L}_{\text{SR}} + w_{\text{Recons}}\mathcal{L}_{\text{Recons}}$.

The last step in describing our algorithm is to define $r_{\text{int}}(S_t;\theta^-)$, the intrinsic reward we use to encourage exploration. We choose the exploration bonus to be the inverse of the $\ell_2$-norm of the vector of successor features of the current state, that is,

$$r_{\text{int}}(S_t;\theta^-) = \frac{1}{||\psi(S_t;\theta^-)||_2},$$

where $\psi(S_t;\theta^-)$ denotes the successor features of state $S_t$ parametrized by $\theta^-$. The exploration bonus comes from the same intuition presented in the previous section, but instead of penalizing the agent with the norm of the SR we make $r_{\text{int}}(S_t;\theta^-)$ into a bonus (we observed in preliminary experiments not discussed here that DQN performs better when dealing with positive rewards). Moreover, instead of using the $\ell_1$-norm we use the $\ell_2$-norm of the SR since our features have unit length in $\ell_2$ (whereas the successor probabilities in the tabular-case have unit length in $\ell_1$).

Finally, we initialize our network the same way Oh et al. (2015) does. We use Xavier initialization (Glorot & Bengio, 2010) in all layers except the fully connected layers around the element-wise multiplication denoted by $\otimes$, which are initialized uniformly with values between $-0.1$ and $0.1$.

Table 2: Performance of the proposed algorithm, $\text{DQN}_e^{\text{MMC}}$+SR, compared to various agents on the "hard exploration" subset of Atari 2600 games. The DQN results reported are from Machado et al. (2018a) while the $\text{DQN}^{\text{MMC}}$+CTS and $\text{DQN}^{\text{MMC}}$+PixelCNN results were extracted from the learning curves available in Ostrovski et al.'s (2017) work. $\text{DQN}_e^{\text{MMC}}$ denotes another baseline used in the comparison. When available, standard deviations are reported between parentheses. See text for details.

| | DQN | | $\text{DQN}^{\text{MMC}}$+CTS | $\text{DQN}^{\text{MMC}}$+PixelCNN | $\text{DQN}_e^{\text{MMC}}$ | | $\text{DQN}_e^{\text{MMC}}$+SR | |
|---|---|---|---|---|---|---|---|---|
| FREEWAY | 32.4 | (0.3) | 29.2 | 29.4 | 29.5 | (0.1) | 29.5 | (0.1) |
| GRAVITAR | 118.5 | (22.0) | 199.8 | 275.4 | 1078.3 | (254.1) | 430.3 | (109.4) |
| MONT. REV. | 0.0 | (0.0) | 2941.9 | 1671.7 | 0.0 | (0.0) | 1778.6 | (903.6) |
| PRIVATE EYE | 1447.4 | (2,567.9) | 32.8 | 14386.0 | 113.4 | (42.3) | 99.1 | (1.8) |
| SOLARIS | 783.4 | (55.3) | 1147.1 | 2279.4 | 2244.6 | (378.8) | 2155.7 | (398.3) |
| VENTURE | 4.4 | (5.4) | 0.0 | 856.2 | 1220.1 | (51.0) | 1241.8 | (236.0) |

## 5 EMPIRICAL EVALUATION OF EXPLORATION IN DEEP RL

We evaluated our algorithm on the Arcade Learning Environment (Bellemare et al., 2013). Following Bellemare et al.'s (2016) taxonomy, we evaluated our algorithm in the Atari 2600 games with sparse rewards that pose hard exploration problems. They are: FREEWAY, GRAVITAR, MONTEZUMA'S REVENGE, PRIVATE EYE, SOLARIS, and VENTURE.[3]

We followed the evaluation protocol proposed by Machado et al. (2018a). We used MONTEZUMA'S REVENGE to tune our parameters (training set). The reported results are the average over 10 seeds after 100 million frames. We evaluated our agents in the stochastic setting (sticky actions, $\varsigma = 0.25$) using a frame skip of 5 with the full action set ($|\mathcal{A}| = 18$). The agent learns from raw pixels, that is, it uses the game screen as input.

Our results were obtained with the algorithm described in Section 4. We set $\beta = 0.025$ after a rough sweep over values in the game MONTEZUMA'S REVENGE. We annealed $\epsilon$ in DQN's $\epsilon$-greedy exploration over the first million steps, starting at $1.0$ and stopping at $0.1$ as done by Bellemare et al. (2016). We trained the network with RMSprop with a step-size of $0.00025$, an $\epsilon$ value of $0.01$, and a decay of $0.95$, which are the standard parameters for training DQN (Mnih et al., 2015). The discount factor, $\gamma$, is set to $0.99$ and $w_{\text{TD}} = 1$, $w_{\text{SR}} = 1000$, $w_{\text{Recons}} = 0.001$. The weights $w_{\text{TD}}$, $w_{\text{SR}}$, and $w_{\text{Recons}}$ were set so that the loss functions would be roughly the same scale. All other parameters are the same as those used by Mnih et al. (2015).

Table 2 summarizes the results after 100 million frames. The performance of other algorithms is also provided for reference. Notice we are reporting *learning performance* for all algorithms instead of the maximum scores achieved by the algorithm. We use the superscript $^{\text{MMC}}$ to distinguish between the algorithms that use MMC from those that do not. When comparing our algorithm, $\text{DQN}_e^{\text{MMC}}$+SR, to DQN we can see how much our approach improves over the most traditional baseline. By comparing our algorithm's performance to $\text{DQN}^{\text{MMC}}$+CTS (Bellemare et al., 2016) and $\text{DQN}^{\text{MMC}}$+PixelCNN (Ostrovski et al., 2017) we compare our algorithm to established baselines for exploration. As highlighted in Section 4, the parameters of the network we used are different from those used in the traditional DQN network, so we also compared the performance of our algorithm to the performance of the same network our algorithm uses but without the additional modules (next state prediction and successor representation) by setting $w_{\text{SR}} = w_{\text{Recons}} = 0$ and without the intrinsic reward bonus by setting $\beta = 0.0$. The column labeled $\text{DQN}_e^{\text{MMC}}$ contains the results for this baseline. This comparison allows us to explicitly quantify the improvement provided by the proposed exploration bonus. The learning curves of these algorithms, their performance after different amounts of experience, and additional results analyzing, for example, the impact of the introduced auxiliary task, are available in the Appendix.

We can clearly see that our algorithm achieves scores much higher than those achieved by DQN, which struggles in games that pose hard exploration problems. Moreover, by comparing $\text{DQN}_e^{\text{MMC}}$+SR to $\text{DQN}_e^{\text{MMC}}$ we can see that the provided exploration bonus has a big impact in the game MONTEZUMA'S REVENGE, which is probably known as the hardest game among those we used in our evaluation. Interestingly, the change in architecture and the use of MMC leads to a big improvement in games such as GRAVITAR and VENTURE, which we cannot fully explain. However, notice that the change

---

[3]The code used to generate these results is available at: `https://github.com/ommitted/blind`.

in architecture does not have any effect in MONTEZUMA'S REVENGE. The proposed exploration bonus seems to be essential in this game. Finally, we also compared our algorithm to DQN[MMC]+CTS and DQN[MMC]+PixelCNN. We can observe that, on average, it performs as well as these algorithms, but instead of requiring a density model it requires the SR, which is already defined for every problem since it is a component of the value function estimates, as discussed in Section 2.

## 6 RELATED WORK

There are multiple algorithms in the tabular, model-based case with guarantees about their performance (e.g., Brafman & Tennenholtz, 2002; Kearns & Singh, 2002; Strehl & Littman, 2008; Osband et al., 2016). RiverSwim and SixArms are domains traditionally used when evaluating these algorithms. In this paper we have given evidence that our algorithm performs as well as some of these algorithms with theoretical guarantees. Among these algorithms, R-MAX seems the closest approach to ours. As with R-MAX, the algorithm we presented in Section 3 augments the state-space with an imaginary state and encourages the agent to visit that state, implicitly reducing the algorithm's uncertainty in the state-space. However, R-MAX deletes the transition to this imaginary state once a state has been visited a given number of times. Ours lets the probability of visiting this imaginary state vanish with additional visitations. Moreover, notice that it is not clear how to apply these traditional algorithms such as R-MAX and E$^3$ to large domains where function approximation is required.

Conversely, there are not many model-free approaches with proven sample-complexity bounds (e.g., Strehl et al., 2006), but there are multiple model-free algorithms for exploration that actually work in large domains (e.g., Stadie et al., 2015; Bellemare et al., 2016; Ostrovski et al., 2017; Plappert et al., 2018). Among these algorithms, the use of pseudo-counts through density models is the closest to ours (Bellemare et al., 2016; Ostrovski et al., 2017). Inspired by those papers we used the mixed Monte-Carlo return as a target in the update rule. In Section 5 we have shown that our algorithm performs generally as well as these approaches without requiring a density model. Importantly, Martin et al. (2017) had already shown that counting activations of fixed, handcrafted features in Atari 2600 games leads to good exploration behavior. Nevertheless, by using the SSR we are not only counting *learned* features but we are also implicitly capturing the induced transition dynamics.

Finally, the SR has already been used in the context of exploration. However, it was used to help the agent learn how to act in a higher level of abstraction in order to navigate through the state space faster (Machado et al., 2017; 2018b). Such an approach has led to promising results in the tabular case but only anecdotal evidence about its scalability has been provided when the idea was applied to large domains such as Atari 2600 games. Importantly, the work developed by Machado et al. (2018b), Kulkarni et al. (2016) and Oh et al. (2015) are the main motivation for the neural network architecture presented here. Oh et al. (2015) have shown how one can predict the next screen given the current observation and action (our auxiliary task), while Machado et al. (2018b) and Kulkarni et al. (2016) have proposed different architectures for learning the successor representation from raw pixels.

## 7 CONCLUSION

RL algorithms tend to have high sample complexity, which often prevents them from being used in the real-world. Poor exploration strategies is one of the main reasons for this high sample-complexity. Despite all of its shortcomings, uniform random exploration is, to date, the most commonly used approach for exploration. This is mainly due to the fact that most approaches for tackling the exploration problem still rely on domain-specific knowledge (e.g., density models, handcrafted features), or on having an agent learn a perfect model of the environment. In this paper we introduced a general method for exploration in RL that implicitly counts state (or feature) visitation in order to guide the exploration process. It is compatible to representation learning and the idea can also be adapted to be applied to large domains.

This result opens up multiple possibilities for future work. Based on the results presented in Section 3, for example, we conjecture that the substochastic successor representation can be actually used to generate algorithms with PAC-MDP bounds. Investigating to what extent different auxiliary tasks impact the algorithm's performance, and whether simpler tasks such as predicting feature activations or parts of the input (Jaderberg et al., 2017) are effective is also worth studying. Finally, it might be interesting to further investigate the connection between representation learning and exploration, since it is also known that better representations can lead to faster exploration (Jiang et al., 2017).

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

# Supplemental Material
# Count-Based Exploration with the Successor Representation

This supplementary material contains details omitted from the main text due to space constraints. The list of contents is below:

- Pseudo-code of the model-based algorithm discussed in Section 3;
- Description of RiverSwim and SixArms, the tabular domains we used in our evaluation;
- Learning curves of $DQN_e$ and $DQN_e^{MMC}+SR$ and their performance after different amounts of experience in the Atari 2600 games used for evaluation;
- Results of additional experiments designed to evaluate the role of the auxiliary task in the results reported in the paper for ESSR.

## EXPLORATION THROUGH THE SUBSTOCHASTIC SUCCESSOR REPRESENTATION

In the main paper we described our algorithm as a standard model-based algorithm where the agent updates its transition probability model and reward model through Equation 2 and its SSR estimate as in Definition 3.1. The pseudo-code with details about the implementation is presented in Algorithm 1.

---

**Algorithm 1** Exploration through the Substochastic Successor Representation (ESSR)

---

$n(s, s') \leftarrow 0 \qquad \forall s, s' \in \mathcal{S}$
$t(s, a, s') \leftarrow 1 \qquad \forall s, s' \in \mathcal{S}, \forall a \in \mathcal{A}$
$\hat{r}(s, a) \leftarrow 0 \qquad \forall s \in \mathcal{S}, \forall a \in \mathcal{A}$
$\hat{P}(s, a) \leftarrow 1/|\mathcal{S}| \quad \forall s \in \mathcal{S}, \forall a \in \mathcal{A}$
$\tilde{P}(s, s') \leftarrow 0 \qquad \forall s, s' \in \mathcal{S}$
$\pi \leftarrow$ random over $\mathcal{A}$
**while** episode is *not* over **do**
$\quad$ Observe $s \in \mathcal{S}$, take action $a \in \mathcal{A}$ selected according to $\pi(s)$, and observe a reward $R$ and a next state $s' \in \mathcal{S}$
$\quad n(s, s') \leftarrow n(s, s') + 1$
$\quad t(s, a, s') \leftarrow t(s, a, s') + 1$
$\quad n(s) \leftarrow \sum_{x', b} t(s, b, x')$
$\quad n(s, a) \leftarrow \sum_{x'} t(s, a, x')$
$\quad \hat{r}(s, a, s') \leftarrow \frac{(t(s,a,s')-2) \times \hat{r}(s,a,s') + R}{t(s,a,s')-1}$
$\quad$ **for each** state $x' \in \mathcal{S}$ **do**
$\quad\quad \hat{P}(s, a, x') \leftarrow \frac{t(s,a,x')}{n(s,a)}$
$\quad\quad \tilde{P}(s, x') \leftarrow \frac{n(s,x')}{n(s)+1}$
$\quad$ **end for**
$\quad \tilde{\Psi} \leftarrow (I - \gamma\tilde{P})^{-1}$
$\quad r_{\text{int}} \leftarrow -\tilde{\Psi}\mathbf{e}$
$\quad \pi \leftarrow \text{POLICYITERATION}(\hat{P}, \hat{r} + \beta r_{\text{int}})$
**end while**

---

DESCRIPTION OF RIVERSWIM AND SIXARMS

The two domains we used as testbed to evaluate the proposed model-based algorithm with the exploration bonus generated by the substochastic successor representation are shown in Figure 2. These domains are the same used by Strehl & Littman (2008). For SixArms, the agent starts in state 0. For RiverSwim, the agent starts in either state 1 or 2 with equal probability.

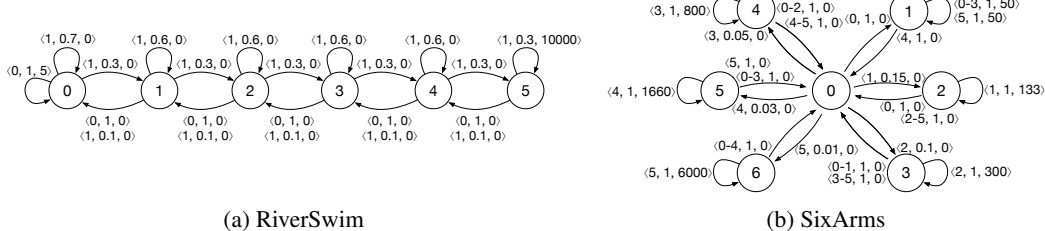

(a) RiverSwim                                        (b) SixArms

Figure 2: Domains used as testbed in the tabular case. The tuples in each transition should be read as ⟨action id, probability, reward⟩. See text for details.

EVALUATION THE IMPACT OF THE AUXILIARY TASK IN ESSR

The algorithm we introduced in the paper, ESSR, relies on a network that estimates the state-action value function, the successor representation, and the next observation to be seen given the agent's current observation and action. While the results depicted in Table 2 allow us to clearly see the benefit of using an exploration bonus derived from the successor representation, they do not inform us about the impact of the auxiliary task in the results. The experiments in this section aim at addressing this issue. We focus on Montezumas Revenge because it is the game where the problem of exploration is maximized, with most algorithms not being able to do anything without an exploration bonus.

The first question we asked was whether the *auxiliary task was necessary* in our algorithm. We evaluated this by dropping the reconstruction module from the network to test whether the initial random noise generated by the successor representation is enough to drive representation learning. It is not. When dropping the auxiliary task, the average performance of this baseline over 4 seeds in MONTEZUMA'S REVENGE after 100 million frames was 100.0 points ($\sigma^2 = 200.0$; min: 0.0, max: 400.0). As comparison, our algorithm obtains 1778.6 points ($\sigma^2 = 903.6$, min: 400.0, max: 2500.0). These results suggest that auxiliary tasks seem to be necessary for our method to perform well.

We also evaluated whether the *auxiliary task was sufficient* to generate the results we observed. To do so we dropped the SR module and set $\beta = 0.0$ to evaluate whether our exploration bonus was actually improving the agent's performance or whether the auxiliary task was doing it. The exploration bonus seems to be essential in our algorithm. When dropping the exploration bonus and the successor representation module, the average performance of this baseline over 4 seeds in MONTEZUMA'S REVENGE after 100 million frames was 398.5 points ($\sigma^2 = 230.1$; min: 0.0, max: 400.0). Again, clearly, the auxiliary task is not a sufficient condition for the performance we report.

The reported results use the same parameters as those reported in the main paper. Learning curves for each individual run are depicted in Figure 3.

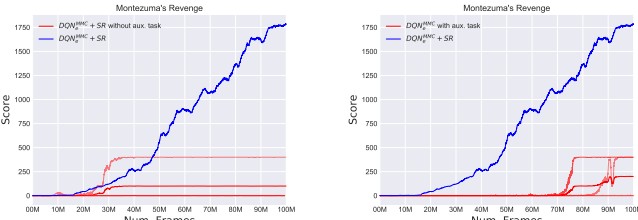

Figure 3: Evaluation of the sufficiency and necessity of the auxiliary task in DQN$_e^{MMC}$+SR. The learning curves are smoothed with a running average computed using a window of size 100.

ADDITIONAL RESULTS FOR DQN$_e^{\text{MMC}}$+SR AND DQN$_e^{\text{MMC}}$ IN THE ATARI 2600 GAMES

As recommended by Machado et al. (2018a), we report the performance of DQN$_e^{\text{MMC}}$+SR and DQN$_e^{\text{MMC}}$ after different amounts of experience (10, 50, and 100 million frames) in Tables 3 and 4.

Finally, Figure 4 depicts the learning curves obtained with the evaluated algorithms in each game. Lighter lines represent individual runs while the solid lines encode the average over the multiple runs.

| Game | 10M frames | | 50M frames | | 100M frames | |
|---|---|---|---|---|---|---|
| FREEWAY | 24.9 | (0.5) | 29.5 | (0.1) | 29.5 | (0.1) |
| GRAVITAR | 244.1 | (23.8) | 326.4 | (53.0) | 430.3 | (109.4) |
| MONT. REVENGE | 2.6 | (7.2) | 563.8 | (465.7) | 1778.6 | (903.6) |
| PRIVATE EYE | 99.2 | (1.2) | 98.5 | (3.3) | 99.1 | (1.8) |
| SOLARIS | 1547.5 | (410.9) | 2036.3 | (339.0) | 2155.7 | (398.3) |
| VENTURE | 26.2 | (22.1) | 942.0 | (423.8) | 1241.8 | (236.0) |

Table 3: Results obtained with DQN$_e^{\text{MMC}}$+SR after different amounts of experience.

| Game | 10M frames | | 50M frames | | 100M frames | |
|---|---|---|---|---|---|---|
| FREEWAY | 25.7 | (1.5) | 29.6 | (0.1) | 29.5 | (0.1) |
| GRAVITAR | 229.9 | (31.3) | 559.3 | (75.9) | 1078.3 | (254.1) |
| MONT. REVENGE | 0.0 | (0.0) | 0.0 | (0.0) | 0.0 | (0.0) |
| PRIVATE EYE | 216.7 | (219.5) | 109.1 | (44.1) | 113.4 | (42.3) |
| SOLARIS | 2230.0 | (322.3) | 2181.5 | (292.9) | 2244.6 | (378.8) |
| VENTURE | 63.8 | (31.3) | 794.1 | (151.9) | 1220.1 | (51.0) |

Table 4: Results obtained with DQN$_e^{\text{MMC}}$ after different amounts of experience.

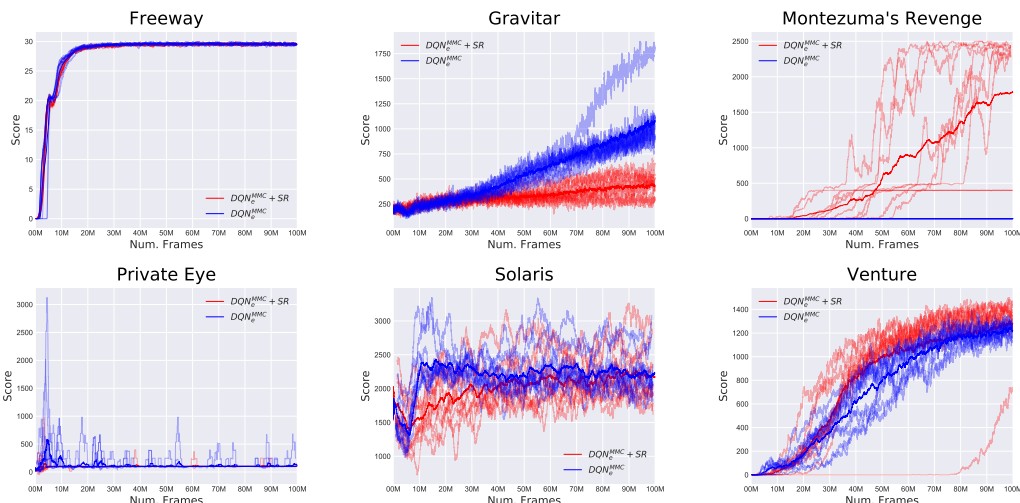

Figure 4: DQN$_e^{\text{MMC}}$+SR and DQN$_e^{\text{MMC}}$ learning curves in the Atari 2600 games used as testbed. The curves are smoothed with a running average computed using a window of size 100.

