# OpenReview forum: "Count-Based Exploration with the Successor Representation"
_ICLR.cc/2019/Conference_

### Official Review · AnonReviewer3 · 2018-11-01
**Theoretically grounded work but lacking convincing results**

**Rating:** 4
**Confidence:** 3

**Review:**

The authors are tackling sample efficiency in the reinforcement learning setting by designing a reward function that encourages exploration. To achieve this they propose you use the successor function which basically counts how often a state has been visited. At first the show this for discrete settings and extend their approach to the continuous state spaces in the Atari 2600 environments.

The paper is well written and the motivation and methods are clear from the beginning.

My biggest concerning is regarding the experimental results of this work. In Table 1 the authors show the results for the tabular games River Swim and Six Arms and copare their approach which they dub ESSR to three methods (E3, R-MAX, MBIE). The numbers in the table indicate that their method ESSR is outperforming E3 and R-MAX on both environments but is itself outperformed by MBIE. The authors don't mention this at al in the respective paragraph nor do the provide a reason as to why this could be case. Also, they neither introduce any of these methods nor do the explain the meaning of the acronyms. Only in the section 6 (of 7) they talk about related works are R-MAX and E3 introduced briefly. But yet again, MBIE is not mentioned.

I have similar concerns about the results presented for the Atari benchmarks. In table 2 the authors compare their method to the classic DQN approach and two more approaches. While their approach outperforms DQN in almost all tasks, this does not hold for the remaining algorithms. Their method is being outperformed in all but one (Venture) task, where they report a higher variance and a small performance boost compared to DQN_e^MMC. Also it is not clear to me where the numbers for the DNQ_e^MMC come from. The authors just say "[...] denotes another baseline used in the comparison". Is this the proposed method of this work but without the successor representation?

In my opinion this work is lacking some clear and convincing results.  Is the main benefit of this method that it does not rely on domain-specific knowledge? If so, then it is not communicated clearly. The authors mention this briefly in the conclusion but provide no further analysis

---

> ### Author Response · Authors · 2018-11-21
> **On the relevance of the presented results**
>
> We thank the reviewer for their feedback. It seems to us that the concerns that were raised were due to some miscommunication regarding how we interpret the results in the paper.
>
> Our experiments in RiverSwim and SixArms were not supposed to show that our algorithm is significantly better than R-MAX, E^3, and/or MBIE. Our experiment was simply to show that the performance of our algorithm is similar to other methods that have PAC-MDP guarantees. This suggests that the proposed idea is promising. This is why we didn't spend too much time discussing the fact that it performs worse than MBIE. We are happy to expand this discussion if you think it is worth.
>
> In a sense, the same argument applies to our Atari results. Our experiments intend to show that our algorithm performs as well as the pseudo-count based methods, but without requiring a density model, which is is a big plus. If we compare the performance of our algorithm to DQN+PixelCNN we see that it outperforms this baseline in 4 out of 6 games. If we compare the performance of our algorithm to DNQ+CTS, DQN+SR outperforms DQN+CTS in 5 out of 6 games! The other baseline, DQN_e^{MMC}, which doesn't use the exploration bonus, is demonstrating something else. It is showing how different parameters of the network can improve the agent's performance. That being said, these different parameters do not help at all in the task with very sparse rewards: Montezuma's Revenge. This shows that the exploration bonus we proposed is indeed effective.
>
> The impression that the numbers we are reporting are “low” comes from the fact that we trained our agents for 100M frames. One of the great papers submitted to ICLR, introducing RND, got scores of 7, 9, and 10 (https://openreview.net/forum?id=H1lJJnR5Ym&noteId=H1lJJnR5Ym). The numbers reported in that paper were obtained after 2 billion frames(!). Unfortunately we do not have the computational power to evaluate our algorithm for that long. However, if we look at the performance of RND at 100M frames, in Figure 7 of the referred paper, our method outperforms RND at 100M frames in Venture and Solaris, it is outperformed by RND in Gravitar, and exhibits comparable performance in Montezuma’s Revenge and Private Eye. It is a well-known fact that most current deep RL algorithms perform better in Atari 2600 games when more frames are provided. We don’t think our paper should be evaluated on our ability to run experiments for 20 times longer. It seems to us that the reviewers assessment in the regard of the reported performance is harsher than what is common at ICLR.
>
> The other concern that was raised was the fact that we do not explain the model-based algorithms used as baseline, nor DQN_e^{MMC}. We didn't explain the model-based algorithms used as baseline in order to be more concise and because they are fairly standard in the field. If the paper is accepted, we are happy to do so in its final version. DQN_e^{MMC} is explained in the fourth paragraph of Section 5. Shortly, it is our algorithm without the exploration bonus.

---

### Official Review · AnonReviewer1 · 2018-11-02
**Not very novel and rather confusing**

**Rating:** 5
**Confidence:** 2

**Review:**

Being familiar but not an expert in reinforcement learning, my review will focus on the overall soundness of the proposed method

Summary:

The authors are interested in the problem of sample efficiency in reinforcement learning, i.e. how to learn a policy achieving good performance (discounted reward) in a RL setting using as little interaction with the environment as possible.
To do this the authors propose to learn a policy in a new environment where the reward has changed: an exploration bonus is added to the reward that should bias the agent towards the least frequently visited states.

The algorithms proposed throughout the manuscript are extensions of a two-part algorithm of the following flavour: 1) An estimate of visitation count is done in an online fashion using a modified version of the successor representation (SR). 2) This estimates parametrizes the exploration bonus of the environment . Both learning algorithms are optimized together.

This initial algorithm is fairly simple in its description and builds on well established ideas in RL. The authors then ‘evaluate the effectiveness of the proposed exploration bonus in a standard model-based algorithm’ against other baselines. They do explain how the model is learned, but not how the policy is optimized.

The remainder of the manuscript applies the same idea to different settings.
For large state spaces, The SR expected visits are learned using TD along with state action value functions. The counts of visitations are replaced by features that are also learned.

Overall, the manuscript is rather confusing.
The SSR theorem is stated (with no real intuition and the actual bounds on n(s) left for the reader to derive). It is not well motivated. Why would we want expected counts and not the discounted version?
Then the remainder of the paper actually makes no use of this theorem, but only use it as a distant inspiration. Tentative connections are made such as TD underestimating SR thus leading to a result more akin to SSR, which is highly speculative. It is also irrelevant since features are learned anyway.

The final proposed architecture has many additions to a simple DQN (the reconstruction + the exploration bonus + the MMC). This makes it difficult to understand what the contribution of the exploration bonus is.
It does not help that results are manually extracted from histograms found in  papers.

Overall, although the intuition is interesting (though not so new).
The overall motivation and structure of this manuscript makes think it does not match the standards of ICLR for publication

---

> ### Author Response · Authors · 2018-11-21
> **Clarifications on the role of the theoretical results in our paper**
>
> We thank the reviewer for their feedback. Regarding our theorem, it intends to provide some intuition of why the norm of the successor representation can be used as an exploration bonus. When we started developing this work, in the tabular case, we observed that the successor representation (SR) could be used to drive exploration when we were computing it online, with temporal-difference (TD) learning. However, this is true only while the SR is being learned, not at its fixed point. At its fixed point the $\ell_1$ norm of the SR is $\sum \gamma^t 1 = 1/(1-\gamma)$ for all states (the SR is a collection of value functions in which a reward of 1 is observed at each time step). Therefore, the converged SR is not able to distinguish between interesting states for exploration because its future state occupancy (the cumulative sum of discounted visitation) is the same for all states. However, when being learned with TD learning, because a reward of $1$ is observed at each time step, there is no variance in the target and the predictions slowly approach the true value of the SR. If pessimistically initialized, the predictions approach the target from below. In this sense, what defines how far a prediction is from its final target is how many times it has been updated in a given state. This is why we used the online updates of TD learning in the function approximation case. We introduce the substochastic SR (SSR) because it is extremely hard to theoretically analyze this transient behavior of TD learning, with only recent papers being able to provide finite-sample complexity bounds for it. The SSR is a clearer way of analyzing the behavior of a predictor that, in the limit, converges to the SR (the $+1$ in the denominator becomes irrelevant) while incorporating the notion of counts before convergence, underestimating the true target but slowly approaching it. Such an analysis is not possible with the closed-form solution of the traditional SR exactly because we are interested in its transient behavior.
>
> Aside from the discussion regarding the usefulness of the theorem along the paper, there were also other comments regarding the clarity of the paper. We address them below:
>
> - We didn't explain how the policy is optimized. We did. See the footnote in page 5. We used policy iteration, a fairly standard algorithm in the field.
> - There's no intuition for the theorem. I'm not sure what the reviewer had in mind here. We do provide an intuition on how the SSR underestimates the SR the same way that TD does while learning. We also discuss how the theorem shows how the SSR counts state visitation.
> - The bounds on n(s) are left for the reader. It is not clear to us what is the source of confusion. Our theorem is explicitly stated in terms of n(s).
> - The use of expected counts and not the discounted version. Traditionally, in the RL literature, expected counts are used. Our work implicitly discounts the counts because the SR has a discount factor. Investigating these different choices is indeed interesting, but this is not the goal of this paper.
> - It is hard to know what the real contributions are on the new architecture. The architecture is not supposed to be the contribution, but the idea of using the norm of the SR as an exploration bonus. The architecture is simply an architecture that instantiates this idea.
> - Results are extracted from histograms. I'm not sure what the reviewer means by this comment. The only results obtained from histograms are some baseline results in Table 1, which is definitely not the main focus of the paper. We explicitly reached out the the authors of the papers we used as baseline in Table 2 in order to get the *exact* numbers they obtained.
> - The intuition is not new. This was a very surprising comment to us and it was not even backed up by references. The other reviewers did acknowledge the novelty of the idea. To the best of our knowledge, using the SR as an exploration bonus has never been done before.

---

### Official Review · AnonReviewer2 · 2018-11-02
**An interesting work but with some flaws**

**Rating:** 5
**Confidence:** 4

**Review:**

This paper proposed a new exploration strategy, based on the successor representation (SR), which can be used as a pseudo bonus in reinforcement learning. The authors also showed the connection between the state visit count and the SR, in the tabular case. Finally, the proposed algorithm had been tested on simulated examples, and several hard exploration Atari domains.

In general, there are some interesting ideas in this paper, while the empirical justification may not be strong enough. My pros and cons are summarized as follows.
Pros:
- The idea of using SR for pseudo count in deep RL is novel.
- Theorem 1 shows the interesting connection between state visit count and the proposed SR.
- The experiments on Atari games show some promise for using SR (but not that much).
Cons:
- There are a few inconsistencies regarding the use of SR. For example, the tabular case used the minus l1 norm as the reward bonus; however, the Atari case instead set the bonus to be the reciprocal of the l2 norm.
- Other than the Montezuma's Revenge, it's difficult to draw the conclusion that using SR can generally lead to better exploration performance, based on the last two columns of Table 2.
- The definition of loss L_{SR} is a bit unclear: Is there something similar to the Bellman equation you can say about SR? I also don't quite understand the motivation for the architecture between \phi and \psi in Figure 1.
- A few small comments/questions are listed as follows.
  1. When discussing the impact of the introduced auxiliary task, it would be more convincing to show the performance of games other than Montezuma's Revenge.
  2. Why is it true that "... because a reward of 1 is observed...", in the second paragraph of Section 4?
  3. What is the value of \tau in the loss L_{TD} on Atari domains?

---

> ### Author Response · Authors · 2018-11-21
> **On the issues raised about the clarity of the paper**
>
> Some issues were raised about the clarity of the paper. We try to clarify the main items below.
>
> - Definition of the loss of the SR: As we discussed in Section 2, the SR can be learned with TD learning, similarly to how a value function is learned. This is exactly what we do with the SR. Thus, our loss is simply motivated by DQN. Finally, there is indeed an equivalent Bellman equation for the SR where the reward is replaced by an indicator function of state visitation (or by a feature vector in the linear function approximation case). We know, since DQN, how to use a deep neural network to learn a value function (minimizing the squared TD error).
>
> - Motivation for the proposed architecture: Our architecture was motivated by related work (Oh et al., 2016; Kulkarni et al., 2016; Machado et al., 2017). It is actually fairly simple. We use an auxiliary task to predict the next screen using the architecture proposed by Oh et al. (2016) and a fully connected layer to predict the SR, which has been shown to work in the past (Kulkarni et al., 2016; Machado et al., 2017). Actually, Machado et al. (2017) basically proposed this architecture, we just added a new head to predict the Q-values, as done by DQN. Our main contribution is not the architecture.
>
> - We refer the reviewer to the equation between equations 3 and 4 in our paper, the one that defines the SR, to explain what we mean by a reward of 1 being observed at each time step. Psi(s,s') is an entry in the matrix. When computing the norm of Psi(s), we are actually looking at a row of the matrix Psi, taking into consideration all future states. I{S_t = s'} is going to be true for only one of the future states, which means that in vector notation, the observed "reward" will be always 1.
>
> - Tau is set to 0.1 in our experiments. Thanks for pointing that out, we will make sure to add this information to the paper.

---

> ### Author Response · Authors · 2018-11-21
> **Clarifying our experimental results and the mismatch between the norms**
>
> We thank the reviewer for their feedback. There is indeed an inconsistency between the use of the norm of the SR in the theorem and in the function approximation case. It is not uncommon though that a theoretically derived result must be adapted to better work when combined with complex function approximators such as neural networks. Shortly, we haven't been able yet to obtain results with the l1 norm that are as good as those we've obtained with the l2 norm. This doesn't change the fact that the theorem does provide an intuition for why the idea we propose works. There is an obvious relationship between the norms as well (e.g., through the Cauchy–Schwarz inequality).
>
> Regarding our results in Atari, we strongly disagree that they are not convincing. Our experiments intend to show that our algorithm performs as well as the pseudo-count based methods, but without requiring a density model, which is is a big plus. If we compare the performance of our algorithm to DQN+PixelCNN we see that it outperforms this baseline in 4 out of 6 games. If we compare the performance of our algorithm to DNQ+CTS, DQN+SR outperforms DQN+CTS in 5 out of 6 games! The other baseline, DQN_e^{MMC}, which doesn't use the exploration bonus, is demonstrating something else. It is showing how different parameters of the network can improve the agent's performance. That being said, these different parameters do not help at all in the task with very sparse rewards: Montezuma's Revenge. This shows that the exploration bonus we proposed is indeed effective.
>
> The impression that the numbers we are reporting are “low” comes from the fact that we trained our agents for 100M frames. One of the great papers submitted to ICLR, introducing RND, got scores of 7, 9, and 10 (https://openreview.net/forum?id=H1lJJnR5Ym&noteId=H1lJJnR5Ym). The numbers reported in that paper were obtained after 2 billion frames(!). Unfortunately we do not have the computational power to evaluate our algorithm for that long. However, if we look at the performance of RND at 100M frames, in Figure 7 of the referred paper, our method outperforms RND at 100M frames in Venture and Solaris, it is outperformed by RND in Gravitar, and exhibits comparable performance in Montezuma’s Revenge and Private Eye. It is a well-known fact that most current deep RL algorithms perform better in Atari 2600 games when more frames are provided. We don’t think our paper should be evaluated on our ability to run experiments for 20 times longer. It seems to us that the reviewers assessment in the regard of the reported performance is harsher than what is common at ICLR.
>
> Regarding the evaluation of the impact of the auxiliary tasks, we did so because the benefit of our proposed exploration bonus is extremely clear in Montezuma's Revenge. This choice was obviously motivated by the cost of running some of these experiments as well (notice we use 10 runs everywhere, which is at least twice the traditional number of runs in the field). If the auxiliary task explained the reported results in Montezuma's Revenge that would mean that our results were fully explained by the auxiliary task and the new network parameters. This is clearly not the case.

---

> > ### Public Comment · (anonymous) · 2018-12-06
> > **Good points**
> >
> > "It is a well-known fact that most current deep RL algorithms perform better in Atari 2600 games when more frames are provided. We don’t think our paper should be evaluated on our ability to run experiments for 20 times longer."
> >
> > Logging in just to say that I couldn't agree more. In sparse reward environments particularly, it's obvious that you'll discover more rewards if you just crank up the training time. Sample efficiency has got to be the most important performance metric on exploration tasks!
> >
> > It's also sad to see this paper, which goes out of its way to provide strong and meaningful baselines, getting significantly worse scores than papers like this one: https://openreview.net/forum?id=Hylyui09tm . Next time you should pick PPO or TRPO as your training algorithm so that you don't have to benchmark against pesky agents that reach higher scores :p

---

> > ### Comment · AnonReviewer2 · 2018-12-07
> > **Comments on the response**
> >
> > Thank you for the response to clarify some of my concerns, which I really appreciate.
> >
> > I am a bit surprised to read the authors' arguments that the low scores "come from the fact that we trained our agents for 100M frames", and the proposed method outperformed the other submission with scores of 7, 9, and 10 at 100M frames.
> >
> > I totally understand that the authors may not have the resource to run as long as the other paper, but it's not convincing for me to raise my score. People can claim their method is better by running a fraction of iterations and achieving higher scores; however, such an unfair comparison implies nothing, from my own perspective. It would be difficult for me to make the prediction that your method will continue to improve, without any sort of guarantees provided. Also, regarding the "well-known fact that most current deep RL algorithms perform better in Atari 2600 games when more frames are provided", it's not consistent with Figure 4 where most of the curves got stabilized around the 100M frames.

---

> > > ### Author Response · Authors · 2018-12-07
> > > **Response**
> > >
> > > Thank you for responding our rebuttal. It was not our intention to suggest that the low scores "come from the fact that we trained our agents for 100M frames". Our sentence was just pushing back on the comment that our results were not convincing. We do think our results are convincing when you compare the performance of our algorithm to similar methods at 100M frames (and this is why we mentioned a paper with high scores). I don't think we, as a community, should only reward papers that had the computational budget to run much longer experiments. Within the budget of 100M frames our results are very convincing.
> > >
> > > About our claim that it is "well-known fact that most current deep RL algorithms perform better in Atari 2600 games when more frames are provided",  we stand behind it. Let us try to clarify why. The learning curves in some of the games we used in our evaluation have sharp increases because the reward function of these games is sparse (see individual runs in Figure 4). The agent needs to "figure out" how to get to the next rewarding state so the learning curve can start increasing again.  The stabilizing flat curves are an artifact of the environment. We refer back to Figure 7 of the RND paper because they ran their experiment for 2 billion frames. In Private Eye, for example,  the learning curve is flat until 1.5 billion frames!
> > >
> > > In Montezuma's Revenge, algorithms that do not promote exploration cannot reach 2500 points even with hundreds of millions of frames. Those that promote exploration sometimes can reach 2500 points in 100M frames. That is what our algorithm does as well. I'm not aware of any algorithm that does better within such short time frame. More frames mean that the agent would have time to figure out how to reach the next rewarding state. We realize this discussion was not clear in the paper and we are happy to make it clearer if the paper is accepted.
> > >
> > > Finally, we do agree with you that we present no guarantees that our method would keep improving at the rate of other method. This is not our goal. Exploration is about reducing the sample complexity of learning algorithms because useful information is collected faster. This is exactly what we evaluate by running up to 100M frames. Maybe it was not clear from our paper that we were comparing ourselves against other baseline results with a small number of frames. We are also more than happy to be more clear about this if our paper is accepted.

---

### Meta-Review · Area_Chair1 · 2018-12-16
**Interesting idea, still some more to show potential**

**Confidence:** 4
**Recommendation:** Reject

**Metareview:**

This paper was on the borderline. I am sympathetic to the authors' point about computational resources. It is helpful to demonstrate performance gains that offer "jump start" performance benefits, as the authors argue. However, the empirical results even on this part are still somewhat mixed-- for example, the proposed approach struggles on Private Eye (doing far worse than DQN) in Table 2. In addition, while it is beneficial to remove the need for training a density model, it would be good to show a place where a density model fails (perhaps because it is so hard to find a good one) compared to their proposed approach.